# The effects of negative social media connotations on subjective wellbeing of an ageing population: A stressor-strain-outcome perspective

Izzal Asnira Zolkepli[1], Rehan Tariq[1]*, Pradeep Isawasan[2], Lalitha Shamugam[1], Hasrina Mustafa[1]

1 School of Communication, Universiti Sains Malaysia, George Town, Pulau Pinang, Malaysia, 2 College of Computing, Informatics and Mathematics, Universiti Teknologi MARA, Perak Branch, Seri Iskandar, Malaysia

* rehantariq@usm.my

**Data Availability Statement:** All data files are available from the Kaggle database (https://www.

## Abstract

In recent years, users' privacy concerns and reluctance to use have posed a challenge for the social media and wellbeing of its users. There is a paucity of research on elderly users' negative connotations of social media and the way these connotations contribute to developing passive behaviour towards social media use, which, in turn, affects subjective wellbeing. To address this research vacuum we employed the stressor-strain-outcome (SSO) approach to describe the evolution of passive social media use behaviour from the perspective of communication overload, complexity, and privacy. We conceptualized subjective wellbeing as a combination of three components–negative feelings, positive feelings, and life satisfaction. Negative and positive feelings were used to derive an overall affect balance score that fluctuates between 'unhappiest possible' and 'happiest possible'. The proposed research framework was empirically validated through 399 valid responses from elderly social media users. Our findings reveal that communication overload and complexity raise privacy concerns among social media users, which leads to passive usage of social media. This passive social media use improved the subjective wellbeing favourably by lowering negative feelings and raising positive feelings and life satisfaction. The findings also revealed that respondents' overall affect balance leans towards positive feelings as a consequence of passive social media use. This study contributes to the field of technostress by illuminating how the SSO perspective aid the comprehension of the way passive social media use influences the subjective wellbeing of its users.

## Introduction

There are two important tendencies that describe the contemporary phase of social change in advanced nations. First, there is the issue of demographic shift, which is associated with a growing ageing population [1]. This shift is an outcome of a low birth rate and declining

kaggle.com/datasets/rehantariq1/subjective-wellbeing-of-ageing-population).

**Funding:** Izzal Asnira Zolkepli Malaysian Ministry of Education under the Fundamental Research Grant Scheme FRGS-MRSA/1/2018/SS09/USM/02/2 The funders had no role in study design, data collection and analysis, decision to publish, or preparation of the manuscript.

**Competing interests:** The authors have declared that no competing interests exist.

proportion of the younger population. Second, there is a shift towards a more "mediatised" way of living that is spurred on by the rapid pace of technological advancement and the penetration of modern information systems (IS) applications. Initially it was perceived that the ageing segment of the population avoided access to these applications and thus benefitted less from such technology in their daily living [2]. However, the ubiquitous presence of Internet technology embedded in smartphones and tablets has also expanded the penetration of modern IS among the ageing population. Social media, virtual communities, and user-generated content constitute the foundation of the modern IS and have developed information-based connectivity among the ageing population [3].

The foregoing situation has prompted further enquiry into the contribution of social media interaction to the ageing population's subjective wellbeing (SWB). Earlier research [see 4, 5] demonstrated that utilising social media directly contributes to the wellbeing of an ageing society as it reduces loneliness, boredom, and social isolation. Although the primary motivation for introducing ageing people to social media was to improve their SWB, it is now crucial to comprehend the possible drawbacks of this trend. Due to its pervasive nature, social media has also sparked discussions among academics about its influence on users' wellbeing resulting from the negative connotations [6]. And, as there is a gap in the literature regarding this, it is worth examining whether social media interaction has any negative implications for the wellbeing of an ageing population. For this purpose, we chose Malaysia, a country with a growing aging population [7] and extensive social media penetration [8]. In Malaysia the percentage of people aged 65 and above increased from 7.0 per cent to 7.4 percent [9] and it is projected to become an ageing nation by 2035 [10].

Using the stressor-strain-outcome (SSO) paradigm as the theoretical ground for this study, this research makes considerable contributions to the existing body of knowledge. First, by concentrating on social media stressors, we broaden the research concerning the effect of the negative connotations of social media on an ageing society. Second, our study elucidates the underlying principle of how SSO variables affect the subjective wellbeing of an ageing population, thereby expanding the potential application of the SSO model. In uncovering the underlying mechanism of SSO we incorporated the recommendations of Fu et al. [11] by including communication overload and complexity as stressors, and replacing social media exhaustion with privacy invasion under strain.

Here, we also introduced a unique link by introducing boredom as a moderator between the stressor and strain in the SSO framework. For the generalisability of the research model in the social media and study population context we incorporated the suggestions of Fu et al. [11] and collected data from those using multiple social media platforms. We also followed the advice of Cao et al. [12] and recruited participants from a non-student population. Finally, the findings of our research can assist scholars and policymakers in designing interventions or methods to mitigate the harm from social media to an ageing population.

## Literature review

### Social media and subjective wellbeing of an ageing population

Wellbeing is categorised into three forms–subjective wellbeing, eudaimonic wellbeing, and social wellbeing [13]. This research focused on subjective wellbeing since it is the most general kind of wellbeing. It is characterised by the presence of positive and negative feelings, and satisfaction with one's life [14]. It is further regarded as one of the most significant outcomes of the ageing process. Willroth et al. [15] found that, in general, SWB is linked to a lower risk of mortality. Lombardo et al. [16], while testing the life satisfaction context of SWB, demonstrated that poor mental health is a stronger predictor of lower life satisfaction in old age than

poor physical health. However, in this era of IS applications, mental health is significantly influenced by the social media [17].

Presently, in their daily lives, a rising number of older adults are adopting social media platforms to communicate with others. Compared to the younger cohorts, older adults have demonstrated the biggest rise in IS usage through their use of the Internet and social media [18]. This is largely due to it being a convenient and economical way for them to maintain social connections, with the added benefits of access to pertinent information, emotional support, and evaluation assistance [19]. This ease in social media use has distinct effects on the lives of an ageing population. On the one hand, social media platforms facilitate their communication with the online global community and the exchange of knowledge. On the other, it eliminates or removes real-world social relationships from their life by allowing them to escape into a virtual world [4].

Numerous studies conducted over the past decade have revealed the overall impact of social media on SWB in old age. Without undermining the importance of earlier research, it needs to be noted that most studies [see 13, 20, 21] on an ageing population in relation to the social media and SWB context have concentrated on positive outcomes, and overlooked or underestimated any possible harmful consequences. According to Marttila et al. [6], greater attention to the negative effects is required, and, accordingly, we proposed stress for an ageing population in this research.

We investigated the effect of negative social media attributes on the propensity to use it through the lens of the Stressor-Strain-Outcome (SSO) paradigm to gain insights into the link between passive outcomes and SWB. This study, grounded on the SSO model and recommendations of Fu et al. [11], proposed a distinctive social media overload, i.e., communication overload, along with complexity, which serve as stressors. These stressors induce internal strain (i.e., privacy invasion) in the presence of a moderator, named boredom, among an ageing population, which, in turn, affects their behavioural outcome (i.e., passive social media use), and thus leads to SWB (i.e., positive feelings, negative feelings, and life satisfaction).

## Stressor-strain-outcome (SSO) model

The SSO model [22] serves as the foundation for our research framework, which describes the stress process and strain of social media, and, specifically, the sense of privacy invasion among the ageing population and its relationship with SWB. Stressors are environmental triggers that are comprehended as troublesome and potentially disruptive [12]. It is the state of imbalance between the demands of a particular circumstance and one's capacity to fulfil them. Strain and outcome describe psychological and behavioural responses to stressors [23]. Strain also transfers the effect of stressors on the outcome variables [22]. SWB denotes people's assessment of their quality of life. This assessment refers to cognitive judgements about accomplishing significant objectives and goals in an individual's life span, as well as the balance of positive and negative feelings [14].

The SSO model was chosen because it provides a comprehensive framework that facilitates the systematic examination of the relationships between the variables under consideration. Notably, the widespread adoption of the SSO model in various professions [see 19, 22] demonstrates its adaptable nature and broad application. In contrast, alternative models, such as the cognitive-behavioural, social skill, or socio-cognitive models [24], are better suited for delving into specific mechanisms or factors related to social media use but lack the ability to encompass the broader association between stressors, strains, and outcomes. In addition, the SSO model allows for the incorporation of mediating and moderating factors within the process [25]. This shaped our study and facilitated the integration of boredom as a potential moderator in the relationship between stressors and privacy invasion.

Previously [see 26–28], the SSO model has primarily been used to shed light on the stress phenomena in the context of the workplace, and has not been researched extensively in the context of an ageing society that is associated with social media. Our research adds to the existing body of literature by integrating the SSO model and the psychological consequences of social media interaction, the moderating role of boredom, and their relevance in influencing the subjective wellbeing of an ageing society.

Communication overload and complexity are used as stressors to explain the shortcomings of social media engagement. Communication overload is an undesirable state that develops when the user's cognitive capability is surpassed by the communication demands from information applications, such as social media [23]. Whereas complexity refers to the level of effort needed to operate the technology [29]. Referring to the strain we focused on psychological strain, which is the psychological state that potentially influences behavioural outcomes via the cognitive process [23]. This might take various forms, and, in this research, privacy invasion is postulated as a strain. Privacy invasion reflects the users' concerns regarding the protection of personal data while interacting via social media applications.

To identify additional factors that can influence the stressor-strain relationship, we introduced moderation, which is novel in SSO literature, and, for this purpose, we utilised boredom as the moderator as it is catalytic in nature [30]. Strange outcomes are developed in the form of passive social media use, which involves browsing and consuming information without leaving comments or attempting to communicate directly with others [31]. From the subjective wellbeing perspective, it is mainly rooted in life satisfaction, which indicates an individual's assessment of the quality of their personal life [14]. However, Diener et al. [32] segmented SWB into three components: life satisfaction, positive feelings, and negative feelings. Negative and positive feelings can also be utilised to calculate an overall affect balance score that fluctuates between unhappiest possible and happiest possible.

Initially, we incorporated the recommendations of Fu et al. [11] and considered positive and negative feelings while investigating passive social media use. Subsequently, we performed an affect balance analysis of respondents separately to conceive the overall leaning of respondents concerning the affect balance scale introduced by Diener et al. [32]. In essence, we utilised the SSO model as an appropriate approach in this study for linking specific stressors, strains, and outcomes with SWB, while emphasising the negative connotations attached to social media and its influence on the internal state and social behaviour of an ageing population.

## Hypotheses development

The research model displayed in Fig 1 illustrates the connection among the study variables developed on the basis of the SSO model and previous literature. At the proximal level, research has revealed that individual factors, such as psychological stressors, are the key determinants of privacy invasion [12]. Referring to this in our research we hypothesised a positive association of two stressors with the strain of privacy invasion.

Communication overload occurs when communication requirements from social media surpass an individual's communication capabilities, thereby disrupting the user's workflow and causing cognitive strain [33]. A user's regular schedule may be disrupted by communication overload, and the resulting distractions may make it hard for them to focus. Basically, social media makes it possible for people to interact with one another whenever and wherever they desire. People use social media to communicate with friends, and frequently check their profiles to make sure they do not miss any messages and can answer immediately when necessary [3]. The pace at which users receive social media messages, comments, and alerts grows in

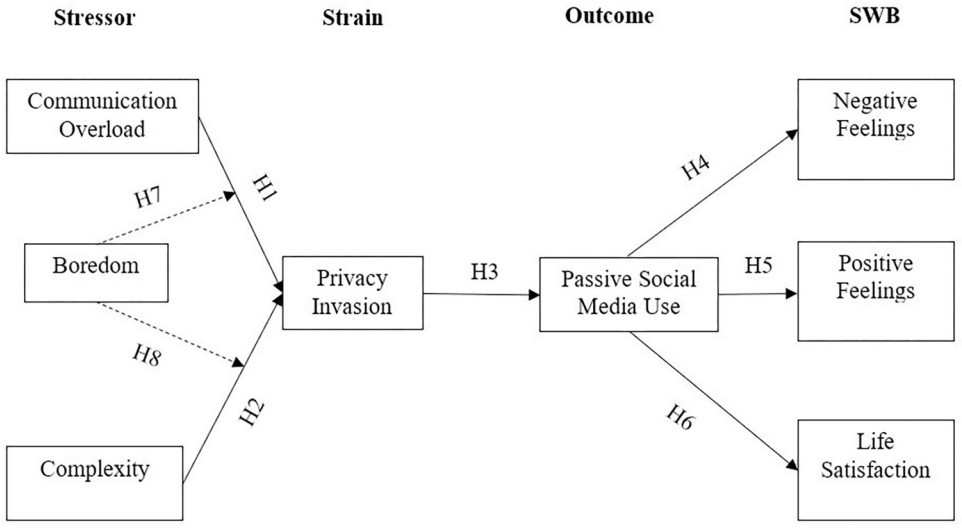

**Fig 1. Conceptual framework.**

proportion to the size of their networks [34]. People's concerns about privacy leaks and related problems are on the rise as they are used to responding to these messages, comments, and alerts with the same pace on several social media platforms while sharing personal information like age, date of birth, and name [35]. Thus, social media becomes a source of privacy invasion. Consequently, we propose that:

H1: Communication overload significantly influences the sense of privacy invasion among older people.

Complexity as a behavioural-based notion, is another barrier to technological adoption. It is best characterised by the degree to which an invention is seen as being somewhat challenging to utilise. It increases the effort to process information [29]. In the social media context, complexity is the degree of effort required to operate social media. High levels of complexity in social media will make it more difficult to use them since it is linked to uncertainty and unpredictability [36]. Furthermore, the confusion arising from the privacy settings associated with social media personal accounts has become an immensely compromising set of tools that facilitate sharing information with third parties [37, 38]. As such a scenario gives rise to a complex psychological response that contributes to the development of privacy concerns [35, 39], we hypothesised that:

H2: Complexity significantly influences the sense of privacy invasion among older people.

In terms of social media use behaviour, there are two possible behavioural consequences for strain: active and passive social media use. Active social media use is associated with lower symptoms of anxiety and depression whereas passive social media use is associated with increased symptoms of anxiety and depression [31]. Previous studies on the behavioural impacts of privacy invasion have primarily resulted in a discontinuity in social media use [31, 35]. The impression that an individual's privacy has been violated as a result of what social media does with their information is pervasive, and has been exacerbated by the infiltration of social media apps [40]. Studies have revealed that privacy is a primary concern for those who frequently use social media [41]. The integration of location-based services in social media platforms make users' particular whereabouts accessible, and, potentially, results in a loss of privacy that creates psychological strain [42], which, in turn, leads to poor performance and low engagement in activities [43]. In the social media environment, once users realise that the

information they provide on social media is being monitored, leaked, or exploited, they may choose to limit or even cease their usage [44]. Accordingly, we propose that:

H3: A sense of privacy invasion significantly influences the intentions of passive social media use among older people.

Research on wellbeing has marked that personal wellbeing may be largely under individuals' control through the activities they choose to engage in and their responses to circumstances in their social surroundings [45]. One of the activities involves how people use social media, in particular, whether they do so actively or passively [46]. Despite previous studies on passive social media use and SWB, further investigation is required to truly understand the relationship between these two variables [47]. Moreover, investigating the relationship between these variables when passive social media use is rooted in the strain of privacy invasion will be a valuable contribution to the existing research. For this purpose, we tested the direct influence of passive social media use on all three components of subjective wellbeing, i.e., positive feelings, negative feelings, and life satisfaction.

Lin et al. [48], while describing the relationship between social media use and depression among American adults, found that passive social media use decreased the symptoms of depression. Similar research proved that the consumption of social media content without engaging in online interaction, i.e., passive social media use, serves as a way of relief for individuals experiencing negative feelings [47]. Moreover, this decline in negative feelings provides room for positive feelings. Thus, we hypothesise that:

H4: Passive social media use significantly contributes to the decline in negative feelings among older people.

H5: Passive social media use significantly contributes to the increase in positive feelings among older people.

Referring to social media use and its association with life satisfaction, Orben et al. [49], while testing the adolescent population, did not find a significant association between them. However, the significance of the impact of passive social media use on wellbeing like life satisfaction has been evidenced [31, 47]. Yue et al. [50], in their research exploring social media browsing and unhealthy social comparison in Wuhan City, China, found that passive use of social media transfers its significant impact on life satisfaction by influencing the wellbeing of citizens. In a similar vein Nguyen and Cheng [51], while testing a moderated mediation model, demonstrated that the way individuals use social media has a significant impact on their life satisfaction. Considering the evidence outlined above we hypothesise that:

H6: Passive social media use significantly contributes to the feelings of life satisfaction among older people.

Boredom has also been gaining attention in various psychological disciplines. As an emotion it can be construed as the engagement in meaningful activities [52], and, generally, it is the outcome of a failed attempt to interact with one's environment; consequently, such dissatisfaction results in emotional stress [53]. It is an unpleasant feeling created by conditions involving a perceived lack of purpose, excitement, and ambition, such as sadness, and the lack of a sense of a purposeful life [43]. Boredom, like other feelings, fluctuates, but prolonged boredom might be damaging to one's health [53]. It is catalytic in nature and couples with an interlocking relationship between psychological outcomes [30]. In conjunction with the aforesaid function of boredom, we propose that boredom may regulate the relationship between stressors and strain in this research. Therefore, the following hypotheses are proposed:

H7: Boredom moderates the relationship between communication overload and the sense of privacy invasion among older people.

H8: Boredom moderates the relationship between complexity and the sense of privacy invasion among older people.

Combining the aforementioned hypotheses, we developed the following framework for this study (Fig 1).

## Methods and measures

### Construct measurement

The question items in the survey were evaluated using a 5-point Likert scale, with 5 representing "strongly disagree" and 1 representing "strongly agree". However, for the measurement items of positive and negative feelings 5 represented "always" and 1 represented "never". The overall affect balance score was measured utilising the technique of Diener et al. [32]–subtracting the negative feelings score from the positive feelings score. We employed a survey research design using an online questionnaire as the instrument for data collection. The targeted population consisted of Malaysian social media users aged 60 and above.

Our survey proceeded with a screening question to confirm that the respondents were social media users and that they were aged 60 and above. The survey included two parts: the first was brief and contained demographic questions, while the second was more extensive and focused on the respondents' behavioural assessment with respect to the research variables. All the items to measure the study constructs were drawn from past research (see S1 Appendix) and validated by three experts in the field of communication. We also conducted a pilot study involving (n = 50) participants to check for any discrepancies. Following the pilot study results and the experts' opinion we revised some items and produced refined and validated measurement items.

### Study setting and data collection

The conceptual framework was tested on the Malaysian population that is predicted to become an ageing population over the next thirteen years. According to the demographic stats, 11.2% of the total Malaysian population comprises ageing people, which is more than 3.5 million. Using the formula of Krejcie and Morgan [54] a sample size of 385 was determined for a population of 3.5 million. In addition, G*power 3.1 with a medium effect size of 0.15 and error 0.05 resulted in 0.99 actual power for the sample of 385. Before the data collection we obtained ethical approval from the human research ethics committee of USM under study protocol code USM/JEPeM/20060292. The study was conducted in 2021 and written informed consent was also obtained from the respondents. A self-administered questionnaire created with the Google form helped us gather the responses.

Using the respondents-driven sampling (RDS) chain referrals approach [55], we aimed to gather data from 385 social media users aged 60 and above. The RDS started with the recruitment of seeds, that is, a label used for early respondents who fulfilled the attributes for the inclusion criteria. After the announcement was made on social media a total of 118 seeds approached the research team who requested their help in recruiting other participants. This process continued until we reached the required number. Our questionnaire received 399 valid responses. The characteristics of the respondents are presented in Table 1.

## Results

Descriptive analysis (see Table 2) was performed on SPSS version 27, and Smart PLS 3.3.9 was utilised to perform the structural equation modelling (SEM) and reveal the relationship among the variables [56]. As the data were obtained from a single source, verification for common method bias (CMB) is essential [57]. The Harman Single Factor test was performed to check the presence of common method bias. The output explained that the first unrotated

**Table 1. Demographic characteristics.**

|  | Description | Frequency | Percentage |
|---|---|---|---|
| **Age** | 60–64 | 249 | 62.4 |
|  | 65–69 | 107 | 26.8 |
|  | 70 and above | 43 | 10.8 |
| **Gender** | Male | 216 | 54 |
|  | Female | 183 | 46 |
| **Marital status** | Married | 332 | 83.2 |
|  | Widowed | 50 | 12.5 |
|  | Single | 10 | 2.5 |
|  | Divorced | 7 | 1.8 |
| **Race** | Malay | 144 | 36.1 |
|  | Chinese | 127 | 31.8 |
|  | Indian | 120 | 30.1 |
|  | Others | 8 | 2 |
| **Education** | Primary | 37 | 9.3 |
|  | Secondary | 117 | 29.3 |
|  | Diploma or the equivalent | 157 | 39.3 |
|  | Bachelor's | 70 | 17.5 |
|  | Master's | 13 | 3.3 |
|  | PhD | 5 | 1.3 |
| **Employment status** | Employed | 28 | 7 |
|  | Retired | 322 | 80.7 |
|  | Self-employed | 43 | 10.8 |
|  | Unable to work | 6 | 1.5 |

factor acquired only 32.12% of the variance in data, which is lower than the threshold value of 50% [58]. Hence, this study did not encounter any problems due to CMB. Furthermore, a stricter alternative technique known as the variance inflation factor (VIF) was utilised to demonstrate the absence of collinearity. According to Hair et al. [59] a VIF greater than 5 indicates a high level of collinearity. However, in the present study it ranged from 1.00 to 1.67, which is below the threshold of 5. Hence, collinearity has no effect on the model in the present study.

## Measurement model

We assessed convergent validity, internal consistency, and discriminant validity to determine the accuracy of the measurement model. For convergent validity, the average variance extracted (AVE) and outer loadings of the indicators were examined, and the findings depicted that both were over the acceptable cut off criterion of 0.50 [56]. Similarly, Cronbach's alpha, composite reliability (CR), and rho_A were all above the specified threshold values of 0.60, 0.70, and 0.70, respectively [60]. Hence, the study variables fulfilled the internal consistency reliability criteria (see Table 2). Discriminant validity was not an issue for our research as the HTMT values were lower than 0.85 [61] (see Table 3).

## Structural model

At first, this research examined the interaction between the stressors and strain. Consistent with our expectations, communication overload ($p < 0.001$, t = 8.902, CI = 0.213; 0.331) and complexity ($p < 0.001$, t = 4.386, CI = 0.109; 0.281) were positively associated with privacy invasion. Afterwards, we evaluated the impact of strain on the behavioural outcome (i.e.,

**Table 2. Constructs' descriptive, convergent validity and internal consistency.**

| Variables | Items | Loadings | rho_A | CR | AVE |
|---|---|---|---|---|---|
| Complexity<br>Mean = 2.027<br>SD = 0.852<br>CA = 0.932 | CMP1 | 0.832 | 0.950 | 0.949 | 0.787 |
| | CMP2 | 0.815 | | | |
| | CMP3 | 0.926 | | | |
| | CMP4 | 0.943 | | | |
| | CMP5 | 0.913 | | | |
| Communication Overload<br>Mean = 3.227<br>SD = 0.850<br>CA = 0.804 | CO1 | 0.583 | 0.968 | 0.864 | 0.621 |
| | CO2 | 0.880 | | | |
| | CO3 | 0.931 | | | |
| | CO4 | 0.711 | | | |
| Boredom<br>Mean = 2.554<br>SD = 0.908<br>CA = 0.889 | BM1 | 0.895 | 0.899 | 0.919 | 0.695 |
| | BM2 | 0.868 | | | |
| | BM3 | 0.748 | | | |
| | BM4 | 0.844 | | | |
| | BM5 | 0.805 | | | |
| Privacy Invasion<br>Mean = 2.399<br>SD = 0.695<br>CA = 0.774 | PIV1 | 0.771 | 0.781 | 0.855 | 0.597 |
| | PIV2 | 0.825 | | | |
| | PIV3 | 0.714 | | | |
| | PIV4 | 0.778 | | | |
| Passive Social Media Use<br>Mean = 3.085<br>SD = 1.085<br>CA = 0.917 | PSMU1 | 0.871 | 0.920 | 0.942 | 0.804 |
| | PSMU2 | 0.954 | | | |
| | PSMU3 | 0.951 | | | |
| | PSMU4 | 0.801 | | | |
| Negative Feelings<br>Mean = 2.959<br>SD = 1.081<br>CA = 0.948 | NF1 | 0.821 | 0.950 | 0.959 | 0.797 |
| | NF2 | 0.903 | | | |
| | NF3 | 0.945 | | | |
| | NF4 | 0.907 | | | |
| | NF5 | 0.866 | | | |
| | NF6 | 0.908 | | | |
| Positive Feelings<br>Mean = 3.471<br>SD = 0.934<br>CA = 0.924 | PF1 | 0.827 | 0.930 | 0.941 | 0.727 |
| | PF2 | 0.910 | | | |
| | PF3 | 0.912 | | | |
| | PF4 | 0.859 | | | |
| | PF5 | 0.808 | | | |
| | PF6 | 0.791 | | | |
| Life Satisfaction<br>Mean = 3.197<br>SD = 0.944<br>CA = 0.871 | LS1 | 0.869 | 0.886 | 0.907 | 0.665 |
| | LS2 | 0.894 | | | |
| | LS3 | 0.881 | | | |
| | LS4 | 0.669 | | | |
| | LS5 | 0.739 | | | |

passive social media use). Again, the result was consistent with our expectations; the sense of privacy invasion was significantly associated with passive social media use ($p < 0.001$, $t = 13.846$, CI = 0.444; 0.592). Next, we tested whether behavioural outcome in the form of passive social media use among older people had an influence on the postulates of subjective wellbeing (i.e., negative feelings, positive feelings, and life satisfaction).

Interestingly, passive social media use demonstrated a significant but negative influence on negative feelings, which depicted an inverse relationship between them ($p < 0.001$, $t = 41.22$,

**Table 3. Discriminant validity HTMT.**

|  | CMP | CO | BM | PIV | PSMU | NF | PF | LS |
|---|---|---|---|---|---|---|---|---|
| CMP |  |  |  |  |  |  |  |  |
| CO | 0.197 |  |  |  |  |  |  |  |
| BM | 0.380 | 0.255 |  |  |  |  |  |  |
| PIV | 0.312 | 0.419 | 0.706 |  |  |  |  |  |
| PSMU | 0.067 | 0.172 | 0.401 | 0.629 |  |  |  |  |
| NF | 0.147 | 0.239 | 0.204 | 0.137 | 0.134 |  |  |  |
| PF | 0.089 | 0.137 | 0.266 | 0.264 | 0.587 | 0.301 |  |  |
| LS | 0.210 | 0.242 | 0.572 | 0.774 | 0.746 | 0.232 | 0.402 |  |

CI = -0.797; -0.725). However, for positive feelings (p < 0.001, t = 13.97, CI = 0.479; 0.633) and life satisfaction (p < 0.001, t = 21.769, CI = 0.596; 0.716) passive social media use demonstrated a significantly positive association. Table 4 shows the results of our structural model analysis.

Adhering to the guidelines prescribed by Hair et al. [59], moderation analysis was performed to ascertain whether boredom weakens or strengthens the relationship between stressors (complexity and communication overload) and strain (privacy invasion). Boredom demonstrated a significant but negative moderation and weakened the relationship between complexity and privacy (p < 0.001, t = 7.703, CI = -0.373; -0.220), while boredom displayed a negative but insignificant moderation (p > 0.50, t = 0.293, CI = -0.072; 0.060) between communication overload and privacy invasion.

We looked at the coefficient of determination ($R^2$ value) to evaluate the model's predictive ability. $R^2$ measures the percentage of variance in the endogenous variables explained collectively by the exogenous variables [59]. The study conducted by Rasoolimanesh et al. [62] categorised the level of acceptance for $R^2$ values in the inner path model for endogenous variables as substantial, moderate, and weak, with corresponding values of 0.67, 0.33, and 0.19, respectively. Our findings revealed that the $R^2$ value for negative feelings ($R^2$ = 0.582) and life satisfaction ($R^2$ = 0.432) demonstrated the moderate predictive power of the model, whereas positive feelings ($R^2$ = 0.31) described weak predictive power. Collectively, the exogenous

**Table 4. Results of the direct and moderating hypotheses.**

| Relationship | Path coefficient (β) | t-value | Confidence interval | | Result |
|---|---|---|---|---|---|
|  |  |  | 2.50% | 97.50% |  |
| CO -> PRI | 0.269*** | 8.902 | 0.213 | 0.331 | Accepted |
| CMP -> PRI | 0.194*** | 4.386 | 0.109 | 0.281 | Accepted |
| PRI -> PSMU | 0.522*** | 13.846 | 0.444 | 0.592 | Accepted |
| PSMU -> NF | -0.763*** | 41.22 | -0.797 | -0.725 | Accepted |
| PSMU -> PF | 0.557** | 13.97 | 0.479 | 0.633 | Accepted |
| PSMU -> LS | 0.658*** | 21.769 | 0.596 | 0.716 | Accepted |
| BOR*CO -> PRI | -0.010 n.s. | 0.293 | -0.072 | 0.06 | Rejected |
| BOR*CMP -> PRI | -0.296*** | 7.703 | -0.373 | -0.220 | Accepted |

*, p < 0.05

**, p < 0.01

***, p < 0.001

n.s., not significant; CO, communication overload; CMP, complexity; BM, Boredom; PIV, privacy invasion; PSMU, passive social media use; NF, negative feelings; PF, positive feelings; LS, life satisfaction.

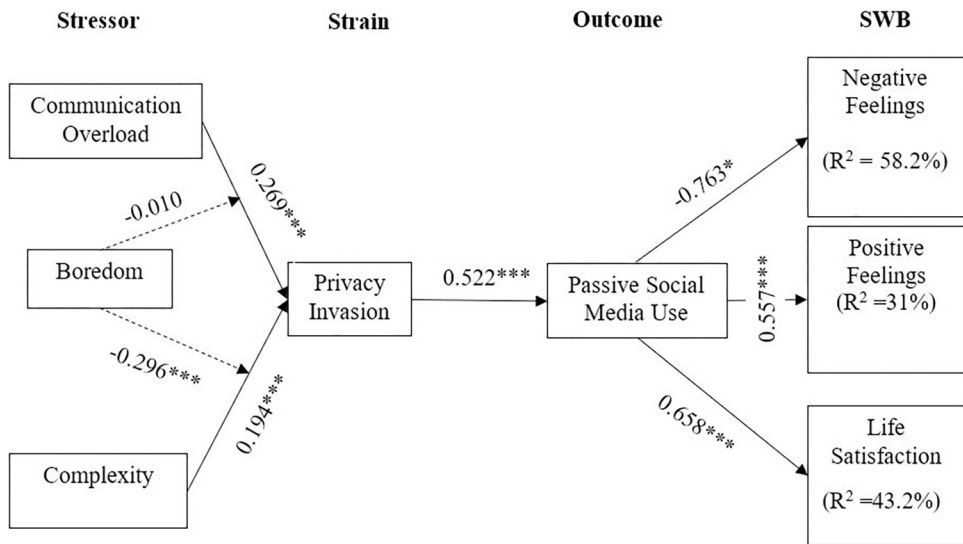

**Fig 2. Conceptual framework with results.**

variables of the study explained 58.2%, 43.2%, and 31% of the variance in negative feelings, life satisfaction, and positive feelings, respectively. The results are shown in Fig 2.

## Affect balance score

Because of the partial independence or distinctiveness of the positive and negative feelings, the positive and negative scales are evaluated independently. The scores on these scales were combined under affect balance for the overall evaluation of the respondents' trend towards negative and positive feelings [32]. Positive and negative feelings each have 6 items that were scored on a scale ranging from "never = 1" to "always = 5". The total positive feelings score can fall anywhere between 6 and 30, and the total negative feelings score can also be anywhere in that range. To combine these under affect balance the sum of the negative feelings score was subtracted from the sum of the positive feelings score. The results ranged from -24 to +24. The majority of the respondents (58.1%) fall on the positive side of the scale, while 35.5% fall on the negative side; 6.4% of the respondents fall right in the middle of the scale depicting a confused state of mind. These findings reveal that respondents' overall affect balance leans toward positive feelings as a consequence of passive social media use.

## Discussion

This study investigated the impact of negative social media connotations on the subjective wellbeing of the ageing population. Although older people are turning to social media platforms, there is little awareness of how consistent interaction on these networks may affect their subjective wellbeing. Fulfilling the recommendation of Cao et al. [12] we have given a substantial demographic shift towards an elderly population and the possible consequences of social media usage for their quality of life, as we felt it was critical to investigate this subject deeper. The SSO framework supported our research goal. This framework comprehends the process that explains the interacting nature of stress and its influence on human behaviour, which is especially important for investigating the complex phenomena investigated in this study.

The research findings, based on the SSO framework, provide novel insights into how communication overload and complexity induce psychological preoccupation as stressors, which then turns into strain in the form of privacy invasion and influences the intention of passive social media use among older people. We also investigated the moderating function of boredom in reducing the impact of complexity on privacy invasion; however, we were unable to identify any considerable effect on the link between communication overload and privacy invasion. Overall, eight hypotheses were investigated in this study, six of which were direct and two of which were indirect in the form of moderation. Except for one indirect hypothesis anticipating the moderating role of boredom between communication overload and privacy invasion, the findings mostly validated the hypothesised relationships. Our research uncovered critical information.

First, both stressors, communication overload and complexity, depicted a significant positive influence on the strain of privacy invasion, supporting H1 and H2. These findings extended the prior research of Fu et al. [11], who suggested examination of the additional dimension of overload, i.e., communication overload, in combination with technostress, such as complexity, on the psychological status of privacy invasion. These findings supplement the results of Li et al. [35] who posited that the considerable communication on social media causes communication overload by demanding that its users respond to this communication while sharing personal information, which, in turn, causes cognitive strain and raises privacy concerns among people. Meanwhile, it is observed that social media exhibits complex personal account privacy settings. Perceptions of illegitimate use of users' shared data and exchange of their private personal data with other parties may give rise to emotions of privacy invasion. Previously, the Cambridge Analytica data scandal, and, more recently, the association of artificial intelligence (AI) with social media sites have already made users more sensitive about the security of their data [63]. The outcomes of our study confirmed this sensitivity by demonstrating that the complexity of social media platforms has a significant positive role in developing the perceptions of privacy invasion among the ageing population of Malaysia.

Second, the perception of privacy invasion exhibited a significant positive influence on passive social media use, which is consistent with past literature [see 35, 43]. When users perceive that their privacy is being compromised, it can cause emotional strain, which, in turn, can manifest as a lack of participation in social media activities and is indicative of the passive usage of social media. Our findings provide credence to this view by showing that users are more likely to reduce or stop using social media when they become aware that the information they share may be monitored and misused. In addition, this research fulfils the recommendation of Fu et al. [11] since we shifted our focus from social media fatigue to privacy invasion when assessing passive social media use.

Third, our findings endorse the overall positive input of passive social media use in SWB. This endorsement is rooted in the significant contribution of the passive use of social media in determining the absence of negative feelings, the presence of positive feelings, and in developing satisfactory feelings towards life. These outcomes are inconsistent with the results of Escobar-Viera et al. [64] and Beyens et al. [34], who found a small aggregate negative effect of passive social media use on youth and teenagers, respectively. This variation in results can be attributed to a different study population and sample size. In this research, the sample is taken from the ageing population of 60 years and above, whereas the aforementioned studies used teenagers and youth. The young population is usually obsessed with social media use and is therefore less likely to have wellbeing after they minimise or abandon social media use.

SWB comprises three types of feelings–negative feelings, positive feelings, and life satisfaction. Because of the partial independence and separability these feelings are measured separately in the measurement and structural model assessment. The negative effect of passive

social media use on negative feelings displays the positive contribution of passive social media use to the SWB of the elderly population of Malaysia. However, the favourable impact of passive social media use on positive feelings and life satisfaction also displays a positive contribution of passive social media use to the SWB of the ageing population of Malaysia.

The study findings related to life satisfaction challenges the results of Orben et al. [49] and Peng et al. [65]. Peng et al. [65] revealed that spending active time on social media venues enhances life satisfaction among students. However, Orben et al. [49] established that social media use pattern is not a strong predictor of life satisfaction across the adolescent population. Our research findings proved the positive contribution of passive social media use in developing life satisfaction among the ageing population in Malaysia. In comparison to the previous research, we proved the varied influence of the pattern of social media use on adolescents and the ageing people.

Fourth, this research extends the literature on the moderating role of boredom between stressors and strain. We observed that boredom, as a moderator, weakened the association between complexity and privacy invasion. As it is well established that boredom is typically the outcome of a failed attempt to engage with one's surroundings, which leads to a lack of vigilance and a decreased level of media attention [66]. Connecting our findings to those of Barkemeyer et al. [66], we are convinced that, in a complex social media environment, boredom promotes a lack of vigilance and a decline in media attention, which leads to the loss of concerns about privacy invasion.

Fifth, while the SSO model has been widely employed in a variety of scenarios, it is relatively new in its application to the specific area of negative social media connotations and subjective wellbeing, particularly, among the ageing population. For instance, Kasim et al. [67] utilised the SSO model to measure the effect of WhatsApp use on the innovative work behaviour of employs. In a similar domain, this model was utilised by Tandon et al. [68] to measure the psychological outcome of fear of missing out among workers on the workplace. Most recently, the trend of employing the SSO model [see 69, 70] to measure the impact of social media and smart phone use on academic performance has also been observed. Our research on the ageing population adds to the current literature on the SSO model and subjective wellbeing by addressing the specific challenges that the elderly population face in the digital age.

In practise, this study provides critical information for social media app developers seeking to avoid subscription attrition. Our findings can assist them in comprehending the causes for social media avoidance and, as a result, reduce passive social media usage. Reducing the complexity of social media app settings, implementing strong privacy measures by customising app features, and implementing anti-hacking capabilities by blocking inbox access to bugs and viruses will also assist app developers in regaining their subscribers. This study also advises the elderly people to restrict their social media friends and utilise the personalised features of social media applications to prevent unwanted messages and comments, which would decrease communication overload and improve privacy.

## Limitations and future directions

There were some limitations in this research, which, if addressed in future research, might change the results. First, the respondents were recruited from one certain age group and one country, so, researchers ought to be mindful to not extrapolate the current findings to other age groups. To overcome this limitation, future research might look at a similar framework with a wider range of samples. Second, the results of the present study are based on data from a cross-sectional survey. When it comes to establishing causal inferences based on connections between variables, cross-sectional surveys have limits. Future research is needed to investigate

the influence of stressors, strain, and their outcomes on subjective wellbeing over time using a longitudinal survey. Along with longitudinal studies, future researchers might use experimental research, which is praised for its high internal validity to establish the causal linkages shown in this study's conceptual framework. An experimental method can be helpful to investigate how specific media platforms, such as Facebook, Twitter, blogs, TikTok, WhatsApp, and YouTube alter attitudes and behaviour in connection to the variables in this study.

## Conclusion

The current demographic and technical transition are linked to the ageing population and the widespread adoption of advanced IS, respectively. Due to the widespread availability of Internet technology incorporated with smartphones, tablets, and social media platforms, IS has increased its penetration among the ageing population. This circumstance demands additional enquiry into the contribution of social media among the ageing people's subjective wellbeing (SWB). Utilising the stressor-strain-outcome (SSO) paradigm, this research goes further by tying together particular stressors, like communication overload and complexity, and strain, such as the impression of privacy invasion. The insights drawn from this study empirically proved that the aforementioned stressors and strain lead to passive social media use. This phenomenon emphasised the negative connotations attached to social media and its effect on the SWB of an ageing population. We expect these extensions can help scholars and practitioners comprehend social media-based IS.

## Supporting information

**S1 Appendix. Measurement items.**
(DOCX)

## Author Contributions

**Conceptualization:** Izzal Asnira Zolkepli, Rehan Tariq, Lalitha Shamugam.

**Data curation:** Rehan Tariq, Pradeep Isawasan, Lalitha Shamugam.

**Formal analysis:** Rehan Tariq, Hasrina Mustafa.

**Funding acquisition:** Izzal Asnira Zolkepli.

**Investigation:** Pradeep Isawasan.

**Methodology:** Rehan Tariq, Pradeep Isawasan, Hasrina Mustafa.

**Project administration:** Izzal Asnira Zolkepli, Hasrina Mustafa.

**Resources:** Lalitha Shamugam.

**Supervision:** Izzal Asnira Zolkepli.

**Validation:** Pradeep Isawasan, Lalitha Shamugam.

**Writing – original draft:** Rehan Tariq.

**Writing – review & editing:** Izzal Asnira Zolkepli, Pradeep Isawasan, Hasrina Mustafa.

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
