## [Decision Letter · Decision Letter 0]

2 Jul 2023

PONE-D-23-13239The Effects of Negative Social Media Connotations on Subjective Wellbeing of Ageing Population: A Stressor-Strain-Outcome PerspectivePLOS ONE

Dear Dr. Tariq,

Thank you for submitting your manuscript to PLOS ONE. After careful consideration, we feel that it has merit but does not fully meet PLOS ONE’s publication criteria as it currently stands. Therefore, we invite you to submit a revised version of the manuscript that addresses the points raised during the review process.

We look forward to receiving your revised manuscript.

Kind regards,

Alejandro Vega-Muñoz, Ph.D.

Academic Editor

PLOS ONE

Reviewers' comments:

Reviewer's Responses to Questions

**Comments to the Author**

1. Is the manuscript technically sound, and do the data support the conclusions?

Reviewer #1: Partly

Reviewer #2: Yes

2. Has the statistical analysis been performed appropriately and rigorously? 

Reviewer #1: Yes

Reviewer #2: Yes

3. Have the authors made all data underlying the findings in their manuscript fully available?

Reviewer #1: Yes

Reviewer #2: Yes

4. Is the manuscript presented in an intelligible fashion and written in standard English?

Reviewer #1: Yes

Reviewer #2: Yes

5. Review Comments to the Author

Reviewer #1: The discussion section needs more articulated explanation and rationale for the study. A detail comparative description of different models may be helpful. The method section need further improvement.

Reviewer #2: These are my comments;

1. There are no clear difference between the paragraph, causing reading to be exhaustive.

2.L173-174 "People use social media to communicate with friends, so they frequently check their profiles to make sure they don't miss"

- do not use short form in academic writing

3. The development of H6 is not enough. Need to add more.

4. L290 "Fortunately, we received 399 responses"- no need to mentioned, already stated above, its repetitive.

5. Common method variance need to be tested using full collinearity test, Harman Single Factor test is no longer valid.

6. There is no further discussion on the R2, explained variance except for reporting.

7. Author only refer to . Fu S, Li H, Liu Y, Pirkkalainen H, Salo M. Social media overload, exhaustion, and use discontinuance: Examining the effects of information overload, system feature overload, and social overload. Inf Process Manag. 2020 Nov;57(6):102307 -- what about other studies?

8. L440-441 "There were some limitations in this research if overcome in future research might change the results"- improve and rephrase this sentence.

9. L450452 "Experimental method can be helpful to investigate how specific media platforms such as Facebook, Twitter, blogs, and WhatsApp alter attitudes and behaviour in connection to the variables in this study."- Did not mention Tik Tok?

10. Suggest to refer to these recent related studies in Malaysia

Kasim, N. M., Fauzi, M. A., Wider, W., & Yusuf, M. F. (2022). Understanding Social Media Usage at Work from the Perspective of Social Capital Theory.

Administrative Sciences, 12(4), 170.

Kasim, N. M., Fauzi, M. A., Yusuf, M. F., & Wider, W. (2022). The Effect of WhatsApp Usage on Employee Innovative Performance at the Workplace: Perspective

from the Stressor–Strain–Outcome Model. Behavioral Sciences, 12(11), 456.

11. Table 4 is wrong, the beta value is not effect size. It is path coefficient.

12. This manuscript require English editing.

6. PLOS authors have the option to publish the peer review history of their article (what does this mean?). If published, this will include your full peer review and any attached files.

Reviewer #1: No

Reviewer #2: **Yes: **Muhammad Ashraf Fauzi

---

## [Author Response · Author response to Decision Letter 0]

5 Aug 2023

Note: The line numbers provided correspond to the file (containing unmarked version of the revised manuscript.

Response: The following modifications have been made to fulfil PLOS ONE’s style requirement.

1. The title has been written in sentence case and capitalized only the first word of the title (L 1,2).

2. Placed the initials of the corresponding author in parentheses after the email address (L11).

2. Please include captions for your Supporting Information files at the end of your manuscript, and update any in-text citations to match accordingly. 

Response: 

1. At the end of the document, provided a caption for the supporting file (L754).

2. Intext citation is updated accordingly (L284).

3. File name for the supporting information file is also updated according to the journal’s guidelines.

Comments of Reviewer #1: The discussion section needs more articulated explanation and rationale for the study. A detail comparative description of different models may be helpful. The method section need further improvement.

Response: 

1. In order to improve the rationale of study in within the discussion section, additional discussion has been incorporated (L393-402). 

2. To improve the articulated information and make a clear expression additional literature has been integrated (L68, 84,396, L477-480). We further enlisted the services of native English-speaking professional proofreader (please refer to the revised manuscript with track changes). 

3. A comprehensive comparative description referring to the different implications of SSO model has been provided (see L475-485)

4. The Method section has been improved with information related to our survey methodology and elucidation of the target population in Malaysia (Refer to the L277-281).

Comments of Reviewer #2:

1. There are no clear difference between the paragraph, causing reading to be exhaustive.

Response: Improved paragraph structure (L58,L98, L135,L150,L160,L352,L459,L467)

2.L173-174 "People use social media to communicate with friends, so they frequently check their profiles to make sure they don't miss"

- do not use short form in academic writing

Response: This particular modification has been executed (L180). However, in order to maintain consistency throughout the document, the manuscript underwent thorough proofreading (please refer to the revised manuscript with track changes).

3. The development of H6 is not enough. Need to add more.

Response:Additional supportive literature has been added to further cultivate the development of H6 (L238-247).

4. L290 "Fortunately, we received 399 responses"- no need to mentioned, already stated above, its repetitive.

Response: The number of respondents, 399, was removed from the previous paragraph and subsequently placed at the end (Line 305), thereby ensuring the coherence and sequential integrity of the data collection procedure.

5. Common method variance need to be tested using full collinearity test, Harman Single Factor test is no longer valid.

Response: To improve the statistical robustness of the data, a full collinearity test (Lines 317-321) was performed, combining a more rigorous statistical approach known as the Variance Inflation Factor (VIF) with the Herman Single factor method.

6. There is no further discussion on the R2, explained variance except for reporting.

Response: Added extensive discussion concerning R2 (Lines 364-374).

7. Author only refer to . Fu S, Li H, Liu Y, Pirkkalainen H, Salo M. Social media overload, exhaustion, and use discontinuance: Examining the effects of information overload, system feature overload, and social overload. Inf Process Manag. 2020 Nov;57(6):102307 -- what about other studies?

Response:We truly appreciate this valuable input, which has substantially improved the article. Our research was primarily guided by Fu and colleagues' recommendations, but in light of the reviewer's direction, we recognised our study's connection with the recommendations made by Cao et al. (12). Furthermore, as seen in L83-84 and L396-402, the inclusion of other studies provided a vital contextual foundation for our research. Likewise, the incorporation of investigations by Kasim et al. (65) and Tandon et al. (66) has resulted in a significant comparative description, thereby shedding light on the diverse implications of the SSO model (Lines 475-485).

8. L440-441 "There were some limitations in this research if overcome in future research might change the results"- improve and rephrase this sentence.

Response:Improved (Lines 496-497) 

9. L450452 "Experimental method can be helpful to investigate how specific media platforms such as Facebook, Twitter, blogs, and WhatsApp alter attitudes and behaviour in connection to the variables in this study."- Did not mention Tik Tok?

Response:Added TikTok and YouTube (Line 508).

10. Suggest to refer to these recent related studies in Malaysia

Kasim, N. M., Fauzi, M. A., Wider, W., & Yusuf, M. F. (2022). Understanding Social Media Usage at Work from the Perspective of Social Capital Theory.

Administrative Sciences, 12(4), 170.

Kasim, N. M., Fauzi, M. A., Yusuf, M. F., & Wider, W. (2022). The Effect of WhatsApp Usage on Employee Innovative Performance at the Workplace: Perspective

from the Stressor–Strain–Outcome Model. Behavioral Sciences, 12(11), 456.

Response: Suggested studies have been added referring to reference number 8 and 65 in intext citation and bibliography.

11. Table 4 is wrong, the beta value is not effect size. It is path coefficient.

Response: Incorporated this correction in Table 4 column 2.

12. This manuscript require English editing.

Response: We incorporated this recommendation (please refer to the revised manuscript with track changes).

---

## [Decision Letter · Decision Letter 1]

23 Oct 2023

PONE-D-23-13239R1The effects of negative social media connotations on subjective wellbeing of an ageing population: A stressor-strain-outcome perspectivePLOS ONE

Dear Dr. Tariq,

Thank you for submitting your manuscript to PLOS ONE. After careful consideration, we feel that it has merit but does not fully meet PLOS ONE’s publication criteria as it currently stands. Therefore, we invite you to submit a revised version of the manuscript that addresses the points raised during the review process.

We look forward to receiving your revised manuscript.

Kind regards,

Alejandro Vega-Muñoz, Ph.D.

Academic Editor

PLOS ONE

Journal Requirements:

Reviewers' comments:

Reviewer's Responses to Questions

**Comments to the Author**

1. If the authors have adequately addressed your comments raised in a previous round of review and you feel that this manuscript is now acceptable for publication, you may indicate that here to bypass the “Comments to the Author” section, enter your conflict of interest statement in the “Confidential to Editor” section, and submit your "Accept" recommendation.

Reviewer #1: (No Response)

Reviewer #2: All comments have been addressed

2. Is the manuscript technically sound, and do the data support the conclusions?

Reviewer #1: Partly

Reviewer #2: Yes

3. Has the statistical analysis been performed appropriately and rigorously? 

Reviewer #1: Yes

Reviewer #2: Yes

4. Have the authors made all data underlying the findings in their manuscript fully available?

Reviewer #1: Yes

Reviewer #2: Yes

5. Is the manuscript presented in an intelligible fashion and written in standard English?

Reviewer #1: Yes

Reviewer #2: Yes

6. Review Comments to the Author

Reviewer #1: Please share how you explored other theories and decided to focus on the stressor-strain-outcome (SSO).

Reviewer #2: Thank you for addressing all the comments in the previous round. The article is now ready for acceptance.

7. PLOS authors have the option to publish the peer review history of their article (what does this mean?). If published, this will include your full peer review and any attached files.

Reviewer #1: No

Reviewer #2: No

---

## [Author Response · Author response to Decision Letter 1]

7 Nov 2023

Reviewer 1: Please share how you explored other theories and decided to focus on the stressor-strain-outcome (SSO).

Response: In order to improve the significance of SSO framework in our study, supportive discussion has been incorporated (L135-144).

---

## [Decision Letter · Decision Letter 2]

27 Dec 2023

The effects of negative social media connotations on subjective wellbeing of an ageing population: A stressor-strain-outcome perspective

PONE-D-23-13239R2

Dear Dr. Tariq,

We’re pleased to inform you that your manuscript has been judged scientifically suitable for publication and will be formally accepted for publication once it meets all outstanding technical requirements.

Kind regards,

Alejandro Vega-Muñoz, Ph.D.

Academic Editor

PLOS ONE

Additional Editor Comments (optional):

This document has been accepted sequentially by 2 independent reviewers.

Reviewers' comments:

Reviewer's Responses to Questions

**Comments to the Author**

1. If the authors have adequately addressed your comments raised in a previous round of review and you feel that this manuscript is now acceptable for publication, you may indicate that here to bypass the “Comments to the Author” section, enter your conflict of interest statement in the “Confidential to Editor” section, and submit your "Accept" recommendation.

Reviewer #1: All comments have been addressed

2. Is the manuscript technically sound, and do the data support the conclusions?

Reviewer #1: Partly

3. Has the statistical analysis been performed appropriately and rigorously? 

Reviewer #1: Yes

4. Have the authors made all data underlying the findings in their manuscript fully available?

Reviewer #1: Yes

5. Is the manuscript presented in an intelligible fashion and written in standard English?

Reviewer #1: Yes

6. Review Comments to the Author

Reviewer #1: I would love to thank the authors for considering my suggestions and addressing them. The paper looks good to me now.

7. PLOS authors have the option to publish the peer review history of their article (what does this mean?). If published, this will include your full peer review and any attached files.

Reviewer #1: **Yes: **TANJIR RASHID SORON

---

## [Editor Report · Acceptance letter]

22 Jan 2024

PONE-D-23-13239R2 

PLOS ONE

Dear Dr. Tariq, 

I'm pleased to inform you that your manuscript has been deemed suitable for publication in PLOS ONE. Congratulations! Your manuscript is now being handed over to our production team.

Kind regards, 

on behalf of

Dr. Alejandro Vega-Muñoz 

Academic Editor

PLOS ONE